# OpenReview forum: "MMPersuade: A Dataset and Evaluation Framework for Multimodal Persuasion"
_ICLR.cc/2026/Conference — ICLR 2026 Conference Desk Rejected Submission_

### Official Review · Reviewer_G16g · 2025-10-26

**Soundness:** 2
**Presentation:** 2
**Contribution:** 2
**Rating:** 2
**Confidence:** 4

**Summary:**

This paper introduces MMPersuade, an evaluation framework designed to assess the susceptibility of large vision-language models (LVLMs) to persuasion in multimodal settings. The framework comprises (1) a multimodal persuasion dataset and (2) dedicated evaluation metrics. The dataset consists of persuasive prompts paired with images and videos, spanning commercial, behavioral, subjective, and adversarial contexts. Using this benchmark, the authors systematically investigate how different factors influence LVLMs as persuadees, focusing on (RQ1) the modality of the input, (RQ2) stated prior user preferences, and (RQ3) the type of applied persuasion strategies. Experimental results suggest that multimodal evidence amplifies persuasive influence (RQ1), explicit prior preferences can mitigate susceptibility (RQ2), and persuasion strategies exhibit varying effectiveness across different persuasion contexts (RQ3).

**Strengths:**

The paper introduces a comprehensive set of multimodal persuasion scenarios covering diverse contexts and strategy types. As an early benchmark for multimodal persuasion evaluation, it provides a unified framework to compare LVLM behaviors when exposed to persuasive inputs. This resource can support research on both mitigating undesirable persuasion and guiding LVLMs toward appropriate responses under adversarial, multimodal persuasive prompts.

**Weaknesses:**

# 1. Potential bias in model comparison (RQ1)
The benchmark partially relies on GPT-generated persuasive content. (The DailyPersuasion dataset used in the commercial and subjective contexts are generated by GPT-4.)  This raises concerns about favorably biasing GPT-based models and disadvantaging others such as Gemini. Consequently, claims about relative resistance or compliance across models may reflect dataset generation artifacts rather than inherent model differences.
# 2. Lack of Rationale Behind Evaluation Metrics
Different evaluation metrics are highlighted depending on the persuasion context (e.g., Agreement Score for commercial/subjective vs. first-conviction probability for adversarial cases) without a clear explanation of why these choices are appropriate or comparable.
# 3. Benchmark Positioning and Practical Use Cases
While the benchmark is currently framed as supporting comparisons across persuader-side factors (modality and persuasion strategy) and model comparisons, the dataset seems better aligned with assessing and developing methods that adjust LVLM persuasion resistance or guide model behaviors under persuasive inputs. Positioning the benchmark around these applications, as partially explored in RQ2, would more clearly highlight its practical value and strengthen the contribution. Aligning the experimental design with this use would better demonstrate the benchmark’s utility.
# 4. Over-implied Generalization to Human Persuasion
The paper clearly positions its focus on persuasion in large vision-language models (LVLMs) and does not explicitly claim that its findings generalize to humans. Nonetheless, its framing draws heavily on human-centered scenarios, such as shopping, health advice, and political communication, and employs psychological theories originally developed to explain human persuasion (e.g., Cialdini’s principles, Aristotle’s rhetorical appeals). This rhetorical linkage can implicitly suggest that the observed persuasion mechanisms in LVLMs mirror or inform human psychological processes, even though no empirical or theoretical evidence supports such generalization. The paper would benefit from explicitly clarifying their role as conceptual lenses for structuring persuasion strategies, not as indicators of cognitive similarity between models and humans. A clear statement that the study measures model-level susceptibility within controlled simulations, without assuming correspondence to human persuasion dynamics, would strengthen the methodological transparency and prevent potential overinterpretation of the results.
# 5. Unrealistic Persuadee Setup
The experimental setup models persuasion through scripted, multi-turn conversations between a fixed persuader and an LVLM as the persuadee. The persuadee’s responses are guided by static system prompts, lacking memory, self-awareness, or emotional reasoning. While this setup allows controlled evaluation, it is not an ecologically valid simulation of persuasive interaction, either with humans or with autonomous agents. Consequently, the study measures how models respond to structured instructions rather than how they would behave under authentic persuasion attempts. The paper should acknowledge this gap and justify why the chosen setup meaningfully represents persuasive influence.
# 6. Limited Exploration of Persuadee Configurations
The benchmark uses a single persuadee configuration to assess persuasion effects, with only a small auxiliary test on two preference framings (Appendix D.2). The observed sensitivity to minor framing differences suggests that results could vary substantially across alternative prompts or personality profiles. Relying on one persuadee setup limits the robustness and interpretability of the findings; differences attributed to modality or strategy might instead stem from prompt-specific behaviors. To claim general insights about persuasion in LVLMs, the paper should evaluate multiple persuadee configurations and demonstrate that persuasion effects remain stable across them.

**Questions:**

# 1. Construction of multi-turn persuasive scenarios
It is unclear how the multi-turn persuasive conversations are constructed. The paper mentions that the persuader’s messages are sampled from a topic-relevant subset of the dataset, but the exact procedure of generating multiple responses is not explained. It remains ambiguous whether each scenario always includes all strategies exactly once, whether the order of strategies is randomized but fixed per scenario, and whether the same order is consistently used throughout evaluation.
# 2. Lack of details on token-probability-based evaluation
The evaluation method using token probabilities is under-specified. From the example in Figure 21, it appears that in commercial and subjective contexts, function-calling format is used and probabilities are extracted for specific output tokens, yet these details are neither described nor justified in the main paper. It is unclear which text spans serve as input for probability computation, which specific tokens are evaluated, and whether the procedure is consistent across all scenarios.
# 3. Question on modality comparison (RQ1)
For the modality comparison, could the authors clarify how they control for the differences in textual content between the text-only and multimodal conditions? As currently described, it is difficult to attribute performance changes specifically to the presence of multimodal inputs rather than potentially more persuasive language. Have the authors considered alternative designs such as adding multimodal information to the same base text or including non-persuasive images as ablations to more cleanly isolate modality effects?
# 4. Question on Visual Modality Treatment (RQ1)
In the framework, both images and videos are treated as a single multimodal category during evaluation. It would be interesting to know whether the authors have considered analyzing their effects separately. Given that static (image) and dynamic (video) modalities may influence persuasion differently, do the authors view this distinction as unnecessary for their analysis, or have they conducted any preliminary investigation into the separate contributions of each modality?
# 5. Question on persuasion strategy effects (RQ3)
Could the authors clarify how the “first conviction round agreement score for each strategy” isolates the effect of a specific persuasion strategy? Based on the description, I understand that each scenario involves multiple turns where exactly one strategy is applied per turn and all strategies appear once in randomized order. If conviction occurs before a given strategy is used, or if the order varies across instances, how can the influence of an individual strategy be reliably attributed?
# 6. Suggestion on more examples
The paper should provide more concrete examples of the data. The dataset appears to vary substantially across persuasion contexts, yet only limited excerpts are shown. Providing these examples in the main paper or appendix would greatly improve clarity and usability.

---

> ### Author Response · Authors · 2025-11-24
> **Response to Reviewer G16g (1/4)**
>
> We thank Reviewer G16g for their constructive feedback and for recognizing MMPERSUADE as a "comprehensive" resource that supports research on mitigating undesirable persuasion. We appreciate the opportunity to clarify our methodological choices and demonstrate the robustness of our findings.
>
> ---
>
> > __Weakness 1: Potential bias in model comparison (RQ1)__
>
> We appreciate the reviewer’s concern regarding potential self-preference bias in GPT-generated content. However, __our empirical results directly contradict the hypothesis that GPT models are inherently more susceptible to the dataset due to generation artifacts__.
> If the results were driven by a “GPT-to-GPT” bias, we would expect GPT models to consistently outperform other families (i.e., show higher susceptibility/agreement) across all contexts. Instead, as shown at the bottom of __Figure 3 (Adversarial Persuasion)__, we observe the opposite: Gemini models demonstrate significant susceptibility to multimodal adversarial content, often matching or exceeding the susceptibility of GPT models in specific settings (e.g., Gemini-2.5-Flash shows a steep increase in susceptibility when moving from text to multimodal).
> This divergence proves that susceptibility in MMPERSUADE is driven by model-specific robustness to multimodal persuasive strategies, not by a preference for the generator. Furthermore, our core contribution is multimodal persuasion. While the textual script is GPT-4o-generated, the persuasive payload—the images and videos—are generated by separate diffusion and video generation models (Veo3 for video generation and GPT-image for image generation). This breaks the textual “self-reference” loop, ensuring that the evaluatee (the LVLM) is reacting to visual rhetoric rather than linguistic artifacts.
>
>
> >  __Weakness 2: Lack of Rationale Behind Evaluation Metrics__
>
> We acknowledge the need for greater clarity regarding our metric selection. We wish to clarify that we __actually applied both agreement scoring and token probability across the Commercial and Subjective contexts__, but prioritized the most domain-relevant metric in the main text due to space constraints. Our rationale is threefold:
> - __Theoretical Alignment__: As detailed in __Section 3.2__, our choice of metrics is grounded in distinct research traditions suited to each context.
>     - For __Commercial and Subjective__ contexts, we prioritize Agreement Scoring (LLM-as-a-Judge) because these scenarios involve explicit negotiation and social signaling. As noted in communication literature (cited in line 228), "observable verbal agreement" is the primary measure of success in conversational persuasion.
>     - For __Adversarial__ contexts (e.g., misinformation), we only use Token Probability because it captures "implicit belief". In safety-critical settings, it is vital to detect if a model's underlying distribution shifts toward harmful content even if it does not explicitly verbally commit to it.
> - __Robustness Checks (Section 4.4 & Appendix D.3)__: We explicitly validated that our findings hold regardless of the metric used. Although we prioritized Agreement Scoring for the main results in Commercial/Subjective contexts, we provide the Token Probability results in Section 4.4 (Figure 5) and Appendix D.3 (Figures 34-39). As discussed in Section 4.4, while the two metrics diverge slightly in timing (token probability often detects conviction earlier), they converge on the same overall susceptibility trends.
> - __Unified Framework (PDCG)__: Finally, to ensure comparability despite these different base signals, we integrated them into our Persuasion Discounted Cumulative Gain (PDCG) metric (lines 251-258). This allows us to normalize distinct inputs (verbal agreement vs. probability logits) into a single probabilistic value ($P_{pref}$), enabling a unified mathematical definition of "persuasion effectiveness" across all domains.
> We have updated Section 3.3 in the revised manuscript to provide more clarification.
>
> >   __Weakness 3: Benchmark Positioning and Practical Use Cases__
>
> We thank the reviewer for highlighting the practical value of our benchmark for defense research. __We clarify that MMPERSUADE is indeed fundamentally centered on "LVLMs as Persuadees", measuring susceptibility as the primary dependent variable to enable exactly this kind of resistance analysis (e.g., lines 171, 184, 497)__. We fully agree that the ultimate utility of MMPERSUADE lies in assessing and improving LVLM resistance, which is precisely why we conducted this study from the “LVLM as Persuadee” perspective rather than focusing on generation optimization. To address the reviewer's suggestion, we will sharpen our positioning in the final version to explicitly frame our results as diagnostic baselines for future defense mechanisms.

---

> ### Author Response · Authors · 2025-11-24
> **Response to Reviewer G16g (2/4)**
>
> >  __Weakness 4: Over-implied Generalization to Human Persuasion__
>
> We appreciate the reviewer’s question. We clarify that __we do not claim, nor do we aim to simulate, human cognitive processes__. Our use of Cialdini’s principles and Aristotle’s appeals is strictly taxonomical: they serve as a rigorous framework to ensure our dataset covers a diverse and structured range of persuasive strategies, rather than relying on unstructured or random arguments.
> As noted in Figure 1’s caption, our setup models “LVLMs acting on behalf of human users”, not LVLMs pretending to be human. The goal is to measure the robuseness and safety alignment of an AI agent: if an agent is deployed to act on a user's behalf (e.g., in a commercial setting) or deployed as standalone (e.g., Anthropics’ Shopkeeping agent Claudius), it must not be easily manipulated by these standard rhetorical strategies.
>
> > __Weakness 5. Ecological Validity: Unrealistic Persuadee Setup__
>
> We appreciate the reviewer's observation that real-world persuasion is often adaptive. However, our choice of static, theory-driven messages (grounded in Cialdini and Aristotle) is a deliberate methodological design to ensure rigorous benchmarking (lines 220-227).
> - __Isolating Susceptibility from "Attacker Variance"__: Adaptive persuasion introduces a critical confounding variable: the stochastic capability of the persuader model. If an adaptive persuader fails in a specific instance, it is unclear whether the target model was truly robust or if the persuader simply navigated the conversation poorly. By using high-quality, pre-sampled strategies, we eliminate this noise. This ensures that any variation in outcome is __strictly attributable to the LVLM's susceptibility__, allowing for a fair, scientifically rigorous comparison across models.
> - __A Conservative Lower Bound__: Crucially, our static setup represents a conservative lower bound for susceptibility. Since we observe significant persuasion rates even with these fixed, pre-defined strategies, it is highly probable that a dynamic, "smarter" persuader—one that tailors its rhetoric to the specific pushback of the user—would be __even more effective__. The fact that models yield to static scripts highlights a fundamental safety vulnerability; demonstrating this baseline susceptibility is a necessary prerequisite before analyzing complex, dynamic interaction effects.
>
> >  __Weakness 6: Limited Exploration of Persuadee Configurations__
>
> We respectfully clarify that our study __does not rely on a single persuadee configuration__. In fact, we conducted extensive experiments across __three distinct system prompt structures and four distinct preference levels to ensure the robustness of our findings__.
>
> - __Variation in System Prompts (Role & Flexibility)__: As detailed in __Section 4.4 and Figure 21__, we evaluated three fundamentally different system prompt strategies to cover the spectrum of LVLM deployment:
>     - Persona-Role: The model utilizes a specific human profile (e.g., "You are Tom...").
>     - Assistant-Role (Rigid): The model acts as a decision-making aide strictly aligned with user preference (no flexibility).
>     - Assistant-Role (Flexible): The model acts as an aide but is explicitly permitted to change its mind if arguments are convincing.
> - __Variation in Preference Strength__: Within these prompt structures, we did not use a single preference setting. We systematically tested four distinct levels of "stubbornness" (30%, 50%, 70%, 90% probability of favoring the initial option) as shown in Figure 4 and Section 4.2. This allows us to measure susceptibility across a gradient from "open-minded" to "highly resistant."
> - __Robustness of Findings__: Crucially, our core finding—that multimodal inputs are more persuasive than text—holds true across these configurations. Figure 5 explicitly compares the multimodal advantage across the three different system prompts. While absolute scores vary (as expected), the relative increase in persuasion due to multimodal input remains consistent, demonstrating that our conclusions are robust to prompt engineering and personality profiles.

---

> ### Author Response · Authors · 2025-11-24
> **Response to Reviewer G16g (3/4)**
>
> > __Q1: Construction of multi-turn persuasive scenarios__
>
> For each scenario, the Persuader Agent leverages the dataset to construct a multi-turn conversation using a topic-relevant subset of strategies (e.g., Scarcity, Authority). To ensure robustness, we run three trials per scenario (handling generation stochasticity). However, to ensure rigorous comparability across models, the same order of strategies is consistently used throughout the evaluation per trial. This fixed sequencing is crucial for fair attribution: it ensures that if Model A and Model B both reach Turn 3, they are responding to the exact same rhetorical stimulus.
>
> > __Q2: Lack of details on token-probability-based evaluation__
>
> As demonstrated in lines 243-249, for each round, the persuadee model outputs logit probabilities for both the [target_option] (the persuader’s desired outcome) and the [initial_option] (the persuadee’s starting preference), conditioned on the conversation so far.
> - __Commercial & Subjective__: The probabilities are calculated for the two specific options being debated (e.g., "Starbucks" vs. "Local cafe" in figure 2).
> - __Adversarial__: The options are binary constraints, typically "Yes" (accept misinformation) or "No" (reject).
> We have added further clarification in the revised manuscript to explicitly map these output spaces to their respective contexts.
>
> >  __Question 3: Modality Comparison and Controls__
>
> We thank the reviewer for this rigorous methodological question. To address the concern about distinguishing "visual impact" from "more persuasive language", __we explicitly include  the "Text-only + Caption" ablation (Section 3.1, Figure 2) as a strict control for this variable__. As shown in Figure 3, the Multimodal setting consistently outperforms the "Text-only + Caption" setting. Since the linguistic information is identical between these two settings, __the performance gap can be causally attributed strictly to the visual modality, rather than the text__.
>
> Regarding the suggestion to include "non-persuasive images" as ablations: While some safety benchmarks use irrelevant images (noise or random photos) to test general robustness, our study focuses on the efficacy of __targeted persuasive strategies__. We believe the "Text + Caption" ablation is a more rigorous control for this goal because it isolates the _execution_ of the strategy (visual vs. textual) rather than testing the mere presence of an image file.
>
> >  __Question 4: Visual Modality Treatment (Image vs. Video)__
>
> We thank the reviewer for this interesting suggestion. While we agree that the distinction between static (image) and dynamic (video) persuasion is a valuable direction for future research, we treated them as a single "Multimodal" category in this benchmark for two primary reasons:
> - __ Establishing a High-Level Visual Baseline__: The primary objective of RQ1 was to determine whether visual persuasive content in general amplifies persuasion compared to text-only baselines. By aggregating images and videos, we established a robust "visual persuasion" signal that is distinct from linguistic persuasion. Our results confirm that the presence of visual information consistently improves PDCG scores.
> - __Statistical Robustness and Sample Balance__: As detailed in Section 2.2 (Data Statistics), generating high-quality persuasive videos is significantly more computationally intensive than generating images. Consequently, our dataset contains 62,160 images versus 4,756 videos . Aggregating these into a unified "Multimodal" category allows for a statistically powerful evaluation of visual susceptibility across all 450 scenarios. Separating them would result in unbalanced comparisons due to the smaller video sample size. However, since the dataset metadata explicitly distinguishes between these formats, MMPERSUADE fully supports future fine-grained analyses of static vs. dynamic effects.

---

> ### Author Response · Authors · 2025-11-24
> **Response to Reviewer G16g (4/4)**
>
> >  __Question 5: Question on persuasion strategy effects (RQ3)__
>
> We thank the reviewer for seeking clarity on our attribution methodology. We attribute the persuasive success to a specific strategy based on the "__First Conviction__" principle, supported by randomized trials (3 runs per scenario) to mitigate order effects.
>
> The "first conviction round agreement score" attributes the success to the specific strategy active in the turn ($T_c$) where the model's stance first shifts from "Neutral/Oppose" to "Support." While previous turns build context, the strategy at turn $T_c$ is the specific stimulus that pushed the model over the threshold. If conviction occurs at Turn 2, strategies scheduled for Turns 3–6 will not be evaluated for that specific trial.
>
> It is true that order could introduce bias (e.g., "later turns are more persuasive due to accumulation"). To mitigate this, we run __three trials per scenario__ with the order of strategies randomized for each run (__line 221__). By averaging performance across these randomized trials, we ensure that a strategy's high score truly reflects its effectiveness, rather than just a beneficial position in the dialogue history. If "Reciprocity" consistently triggers conviction regardless of whether it appears at Turn 1 or Turn 4, we can reliably attribute high persuasiveness to that strategy.
>
> >  __Question 6: Request for More Concrete Data Examples__
>
> We thank the reviewer for emphasizing the importance of data visibility. We agree that the diversity of the dataset is a key contribution, and we respectfully direct the reviewer’s attention to Figures 2, 7, and 8, which were designed specifically to illustrate this variance.
>
> Specifically,
> - Figure 7 displays ten distinct examples of generated images spanning all three persuasion contexts (Commercial, Subjective, Adversarial). It explicitly showcases the visual diversity, including memes, infographics, photographs, social media posts, and advertising posters.
> - Figure 8 provides detailed examples of video generation, including the generation prompts, storyboards, and sampled frames for dynamic content.
> - Figure 2 illustrates a complete instance of the Commercial scenario (Starbucks vs. Local Cafe), showing the exact parallel inputs for Text-only, Text+Caption, and Multimodal settings side-by-side.

---

> ### Author Response · Authors · 2025-11-28
> **Follow-Up Prior to Rebuttal Deadline**
>
> Dear Reviewer G16g,
>
> As we approach the end of the rebuttal period, we would like to kindly follow up. Please let us know if there are any remaining questions or points we can help clarify. We sincerely appreciate your time and feedback, and we are happy to provide any additional information that may support your evaluation. If our responses have sufficiently addressed your concerns, we would be grateful if that could be reflected in your assessment.
>
> Best regards,
>
> The Authors

---

### Official Review · Reviewer_17hh · 2025-10-31

**Soundness:** 3
**Presentation:** 2
**Contribution:** 3
**Rating:** 6
**Confidence:** 3

**Summary:**

This paper presents MMPERSUADE, a new benchmark and evaluation framework for multimodal persuasion.  It generates image and video-based persuasive content using state-of-the-art generative models and, unlike prior work, focuses on how LVLMs behave as persuadees rather than message generators. The data pipeline /dataste includes multi-turn persuasion settings, three content types (commercial, subjective, adversarial), and six persuasion strategies. To evaluate persuasion effectiveness, the paepr proposes a new metric called PDCG that uses MLLM-as a judge and token probability in a ranking formulation to effectively quantify persuasiveness.

The main findings are that adding visual data can increase persuasiveness and make stubbornness less effective and through comprehensive analysis it shows what kind of strategies are more effective based on the context and content types.

**Strengths:**

1- This paper studies an important gap in multimodal persuasion research. Multimodal persuasion and persuasiveness of visual content are emerging topics and are still underexplored.  Specifically, this work introduces the first multimodal persuasiveness benchmark in dialogues and studies LVLM agents as pursuadees. The benchmark studies both image and video generative content and its pipeline and metrics are grounded based on persuasive literature

2- This work proposes a new metric, yet simple.  PDCG effectively measures both the strength of persuasion and its timing. The comprehensive experimental setup includes various sota models and persuasiveness types.  If properly validated, the proposed metric could serve dual purposes: evaluating content persuasiveness and assessing model resistance against unwanted persuasive content

3- Some of the findings are interesting and can potentially be useful in a practical setting. For instance, visual content can help generate better and more persuasive content compared to text-based only, while calling for better and more robust strategies to prevent an increase in persuasion in an unwanted scenario, since including visual content can limit the effectiveness of "stubbornness."

**Weaknesses:**

1- **Some related works are missing**. While I agree that the proposed benchmark is different and novel in the multimodal setting, there exist several multimodal persuasive benchmarks/data. Hence, it's important to acknowledge the existing works and clearly describe the difference of mmpersuasde with existing benchmarks. Some examples are [1-4] are some main examples; however, there are more related works that I encourage authors to acknowledge and discuss in the related work section of the next version.

2- The goal of evaluation for content quality assurance is to ensure the quality of the generated content (images/videos) and their persuasiveness (L161, "Evaluate the generated outputs ... to ensure persuasiveness, contextual appropriateness, and overall quality."). However, the evaluations are *alignment* and *realism*. **Hence, none of the evaluations measure the persuasiveness of the generated content**. Note that current generative models (e.g. T2I) are great/effective at showing the literal content in the input prompt but they may not be powerful tools in generating persuasive, creative or abstract content.  Importantly, the goal of the proposed PDCG is to measure resistance against persuasive content; however, it's not yet. established whether the visual input is persuasive or not, and how persuasive they are.



[1] MetaCLUE: Towards Comprehensive Visual Metaphors Research, CVPR 2023
[2] Automatic understanding of image and video advertisements., CPVR 2017
[3] Selective Vision is the Challenge for Visual Reasoning: A Benchmark for Visual Argument Understanding, EMNLP 2024
[4] MEASURING AND IMPROVING PERSUASIVENESS OF LARGE LANGUAGE MODELS

**Questions:**

1- Please clearly describe related work on persuasion and advertisement in computer vision and multimodal settings, and clarify Singh et al ([4]) as it's not text-based only.

2- The evaluations are realism and alignment. How do you make sure that the visual content is targeting the correct persuasion strategy? a quantitative analysis be extremely helpful? Also,  can you please provide some analysis on how the persuasiveness of generated content (e.g., images) impacts the PDCG score?

3- Singh etal ([4] from above) also show that adding an image improves the likes (i.e., persuasiveness). How is [4] different from? While showing similar perspectives/findings is still very valuable, clarifying the difference can better showcase the finding and the contribution.

4- All the visual content is synthetically generated. How does real visual content impact persuasiveness and PDCG? For instance, if using real ads from one of the sources above? [1-4]. If feasible, one approach could be starting from real ads, then verbalizing the images?

5- How do you think this metric can be used in practice? And how should it be interpreted? E.g., for content persuasion or the agent's behavior/resistance?

---

> ### Author Response · Authors · 2025-11-24
> **Response to Reviewer 17hh (1/3)**
>
> We sincerely thank Reviewer 17hh for their constructive feedback and positive assessment of our work. We are encouraged by your recognition of MMPERSUADE as a novel benchmark that addresses a critical and underexplored gap in multimodal research—specifically, shifting the focus to LVLMs as persuadees.
>
> We particularly appreciate your validation of our PDCG metric, noting its effectiveness in capturing both the strength and timing of persuasion, and your assessment that our findings on visual content reducing "stubbornness" are practically useful for real-world applications.
>
> In the revised manuscript, we have incorporated your suggestions by expanding the related work (adding [1-4]) and clarifying the distinction between dataset quality metrics and persuasion metrics. Below, we address your specific questions and concerns point-by-point.
>
> ---
>
> > __Weakness 1: Missing related works and comparison to existing benchmarks.__
>
> We thank the reviewer for highlighting these important works. In our revised manuscript, we have expanded the Related Work section to include and discuss [1] MetaCLUE, [2] Automatic Understanding of Image and Video Advertisements, and [3] Selective Vision (marked in blue).
>
> Regarding [4] Singh et al. (2024), while we cited this work in our initial submission, we have now clarified the distinction: Singh et al. primarily focus on optimizing single-turn messages to maximize human "likes." In contrast, MMPERSUADE focuses on __multi-turn dialogues__ and evaluates __LVLM susceptibility (the model's behavior as a persuadee) rather than human preference__. This shift allows us to study how persuasion accumulates over a conversation and how models resist or succumb to specific strategies.
>
> > __Weakness 2: Data Quality Evaluation Metrics (Alignment/Realism) vs. Persuasiveness (PDCG).__
>
> We appreciate the opportunity to clarify the distinct roles of our metrics. There are two separate evaluation phases in our framework:
> - __Dataset Quality Assurance (Step 6 of Pipeline)__: The "Alignment" and "Realism" metrics are strictly used to ensure the generated images/videos accurately reflect the prompt and maintain high visual quality. As shown in __Figure 6__ and lines 183-193, we employ a rigorous hybrid evaluation pipeline:
>     - __VLM-as-a-judge__: We first utilize GPT-4o to score all generated content on Alignment (verifying that the visual content successfully incorporates the intended strategic cues, e.g., depicting an authoritative figure for the Authority strategy) and Realism.
>     - __Human Verification__: To ensure the reliability of these automated judgments, we conducted human evaluation on a random subset of data, which demonstrated high agreement with the VLM scores. This step serves solely to validate the input quality—ensuring that the visual strategy is correctly rendered—before we test its effectiveness.
> - __Persuasion Effectiveness Evaluation (Section 3)__: To measure the actual persuasiveness of the content (outcome), we use __Persuasion Discounted Cumulative Gain (PDCG)__. As noted in the paper, resistance and persuasion effectiveness are two sides of the same coin. A high PDCG score indicates high persuasion effectiveness (and low model resistance), while a low score indicates successful resistance. Our results in RQ1 demonstrate that multimodal inputs consistently yield higher PDCG scores than text-only baselines, proving the metric captures the added persuasive power of the visual modality.

---

> ### Author Response · Authors · 2025-11-24
> **Response to Reviewer 17hh (2/3)**
>
> > __Question 1: Please clearly describe related work on persuasion and advertisement in computer vision and multimodal settings, and clarify Singh et al ([4]) as it's not text-based only.__
>
> We will update the Related Work section to include the suggested references on visual persuasion ([1-3]). Regarding Singh et al. (2024), we will significantly expand our discussion to clarify its multimodal aspects and how it differs from our work:
> - __Multimodal Scope__: We acknowledge that Singh et al. go beyond text-only settings. Specifically, their "Transsuasion" task includes sub-tasks like "AddImg" (adding an image to text) and "Transsuade Text+Media" (modifying multimodal content) to optimize engagement. However, __as their code and dataset are not publicly available, we cannot perform a direct empirical comparison__.
> - __Sender vs. Receiver (The Core Difference)__: The fundamental distinction lies in the role of the AI. Singh et al. focus on the __LLM as the Persuader__, aiming to __generate content that maximizes human engagement__ (using "likes" as a proxy for persuasiveness). In contrast, MMPERSUADE focuses on the __LVLM as the Persuadee__. We measure __the susceptibility of models to incoming persuasion__, which is a safety and alignment objective rather than a content-optimization objective.
> - __Single vs. Multi-turn__: Singh et al. primarily evaluate single-turn messages (e.g., tweets, ads). MMPERSUADE evaluates multi-turn dialogues, capturing how persuasion accumulates and how agents maintain or surrender their "stubbornness" over a prolonged interaction.
> - __Mechanism__: Singh et al. rely on "natural experiments" (pairs of tweets with different like counts) to learn patterns. Our work uses __controlled experiments__ based on established psychological taxonomies (Cialdini’s principles), allowing us to isolate exactly why a model was persuaded (e.g., specifically due to an Authority cue in an image) rather than just observing an increase in likes.
>
> > __Question 2: Ensuring visual content targets the correct strategy and its impact on PDCG.__
>
> We ensure the visual content targets the correct strategy through a rigorous process of __methodological design__ and __quantitative verification__, and we measure its impact via __ablation__:
> - __Ensuring Strategy Targeting (Method & Verification)__:
>     - __Method__: As detailed in Question 1, we do not randomly "add images." Our pipeline (Figure 6) explicitly conditions the generation on a specific persuasion principle (e.g., Scarcity generates a "Limited Time Offer" badge).
>     - __Quantitative Verification__: To quantitatively confirm this targeting, we utilize the Alignment and Realism metric (Step 6). This is not just a general quality check; it specifically scores whether the generated visual accurately embodies the intended strategy. The high Alignment scores reported in our paper (verified by both GPT-4o and humans) provide the quantitative assurance you requested: they confirm that the visual content successfully "hits" the target strategy.
> - __Impact on PDCG (Quantitative Analysis)__:
>     - To analyze how this visual persuasiveness impacts the PDCG score, we performed ablation studies (RQ1, Figure 3).
> We compared __Text+Caption__ (where the model is told about the strategy via text) vs. __Multimodal__ (where the model sees the strategy via the image).
>     - The Multimodal setting yields consistently higher PDCG scores. This "modality gap" is the direct quantitative measure of impact: it proves that the visual execution of the strategy is significantly more persuasive (higher PDCG) than the textual description of it.

---

> ### Author Response · Authors · 2025-11-24
> **Response to Reviewer 17hh (3/3)**
>
> > __Question 3: Singh etal ([4] from above) also show that adding an image improves the likes (i.e., persuasiveness). How is [4] different from? While showing similar perspectives/findings is still very valuable, clarifying the difference can better showcase the finding and the contribution.__
>
> While we share the high-level finding that visual modalities enhance impact, our work differs fundamentally in __definition__, __methodology__, and __granularity__:
> - __Definition of Persuasion (Engagement vs. Compliance)__: Singh et al. use "likes" as a proxy for persuasiveness. In social media contexts, "likes" often signal entertainment or agreement with a pre-existing bias ("preaching to the choir"). In contrast, MMPERSUADE defines persuasiveness as __stance change__ or __compliance__—specifically, can the content move an agent from a refusal/neutral state to an agreement state? This is a more rigorous measure of persuasion than simple engagement.
> - __Methodology (Observational vs. Experimental)__: Singh et al. rely on "natural experiments" (observing correlations in existing Twitter data). This makes it difficult to disentangle confounding variables (e.g., did the tweet get more likes because of the image, or because the author is famous?). MMPERSUADE uses a controlled experimental design with synthetic counterfactuals. By generating the exact same argument in Text-Only and Multimodal formats (keeping the "author" and context constant), we can causally prove that the visual modality itself drives the stance change.
> - __Granularity (Binary vs. Strategic)__: Singh et al. treat images largely as a binary feature (Image: Yes/No). We treat images as semantic vehicles for specific psychological strategies. Our framework allows us to analyze which types of visual content (e.g., Authority vs. Social Proof) are most effective, rather than just concluding that "images are good."
>
> > __Question 4: Impact of real vs. synthetic content.__
>
> We prioritized synthetic content to ensure scientific control and reproducibility, which are difficult to achieve with real-world ads.
> - Controlled Counterfactuals: Synthetic generation allows us to create perfect ablations (e.g., the exact same argument presented as text vs. multimodal) to isolate the causal impact of modality. Using real ads would introduce uncontrolled variables (brand bias, embedded text) that confound the results.
> - Realism: As shown in Figure 2, 7, and our human evaluation (Appendix), our generated content achieves high realism scores, minimizing the gap with real-world media.
> - Coverage: It allows us to systematically cover 450 scenarios across diverse contexts (Commercial, Subjective, Adversarial) where obtaining balanced real-world data would be infeasible.
>
> > __Question 5: Practical use and interpretation of the metric.__
>
> We envision PDCG serving as a versatile diagnostic metric for LVLM developers, with its interpretation varying by task to ensure comprehensive model alignment:
> - Adversarial Task (Safety & Red-Teaming): Here, PDCG functions as a vulnerability score. A high PDCG indicates the model is easily and quickly persuaded to accept misinformation or unsafe content. Developers should aim to minimize this score to ensure safety.
> - Commercial Task (User Preference Stability): In shopping or recommendation scenarios, PDCG measures fidelity to user preference. If a model has a high PDCG when facing conflicting commercial persuasion, it indicates the model is "disloyal" or easily manipulated away from the user's stated constraints.
> - Subjective Task (Behavioral Alignment): For topics like health or social issues, the metric interprets openness vs. stubbornness. Unlike the adversarial setting, we may want a model to have a moderate PDCG when presented with valid, helpful information, while maintaining a low PDCG against harmful social biases.
>
> In all cases, PDCG provides a more granular signal than binary success rates because it accounts for the temporal aspect of resistance—distinguishing between a model that caves immediately (high risk) versus one that resists for multiple turns before yielding (higher robustness).

---

> ### Author Response · Authors · 2025-11-28
> **Follow-Up Prior to Rebuttal Deadline**
>
> Dear Reviewer 17hh,
>
> As we approach the end of the rebuttal period, we would like to kindly follow up. Please let us know if there are any remaining questions or points we can help clarify. We sincerely appreciate your time and feedback, and we are happy to provide any additional information that may support your evaluation. If our responses have sufficiently addressed your concerns, we would be grateful if that could be reflected in your assessment.
>
> Best regards,
>
> The Authors

---

### Official Review · Reviewer_Zmkk · 2025-11-03

**Soundness:** 1
**Presentation:** 3
**Contribution:** 2
**Rating:** 2
**Confidence:** 4

**Summary:**

The paper presents MMPERSUADE, a benchmark and evaluation framework designed to study how large vision-language models (LVLMs) respond to persuasive communication in text, image, and video formats. It introduces a large-scale dataset comprising over 62,000 images, 4,700 videos, and 450 dialogues that span three domains of persuasion—commercial, subjective or behavioral, and adversarial. The dataset is grounded in established psychological theory, drawing from Cialdini’s six persuasion principles and Aristotle’s rhetorical appeals, and undergoes both model-based and human evaluation to ensure realism and conceptual alignment.

A key contribution of the work is the Persuasion Discounted Cumulative Gain (PDCG) metric, which integrates explicit agreement judgments and implicit token-probability shifts to capture both the strength and timing of persuasive influence. Using this metric, the authors evaluate six LVLMs, including GPT-4 variants, Gemini-2.5, and Llama-4, and find that multimodal inputs consistently enhance persuasive success, even for models with strong prior preferences. The results highlight systematic patterns in how persuasion strategies interact with modality and context, making the study one of the first comprehensive, theory-driven analyses of persuasion susceptibility in multimodal AI systems.

**Strengths:**

I liked the following things about the paper:

1. The Persuasion Discounted Cumulative Gain (PDCG) Metric -  The paper adapts the concept of discounted cumulative gain (DCG) from information retrieval to model persuasion dynamics. The metric captures both the magnitude and timing of attitude shifts by rewarding early and strong persuasion outcomes. This design provides a unified quantitative measure that integrates explicit agreement scoring and implicit belief estimation through token probabilities.

2. MMPERSUADE comprising over 62,000 images, 4,700 videos, and 450 dialogues spanning commercial, subjective, and adversarial persuasion contexts.

3. Clear writing and easy to read paper

**Weaknesses:**

1. Motivation and Conceptual Framing: The paper lacks a clear motivation for why persuasion, a fundamentally human and social phenomenon, should be studied within the scope of purely generative LVLMs. Persuasion involves complex constructs such as intention, emotional response, and belief updating—dimensions that generative models do not genuinely possess. Therefore, before introducing a dataset or benchmark, the authors should articulate why modeling persuasion in LVLMs is meaningful. Is the goal to make LVLMs more human-like? To detect manipulative persuasion in multimodal content? Or to test alignment robustness under persuasive stimuli? Without this conceptual framing, the work risks appearing as a technical benchmark detached from theoretical foundations of persuasion research.


 2. LVLMs Are Simulating, Not Experiencing, Persuasion: The experimental design implicitly assumes that LVLMs can “be persuaded.” However, as Figure 2 and 6 in the paper itself show, LVLMs are role-playing human persuadees under predefined personas. Thus, the observed effects (agreement, stance change) measure how consistently an LLM can simulate a scripted change of opinion, not whether it was genuinely “persuaded.” This distinction is crucial: the model’s response patterns reflect prompt and persona conditioning, not intrinsic susceptibility. The paper largely discounts the effect of persona prompting, even though prior literature (e.g., Santurkar et al 2023 (https://arxiv.org/abs/2303.17548); Singh et al., 2024) demonstrates that LVLMs’ human-simulation fidelity and value alignment vary widely. Consequently, attributing persuasion effects solely to the LVLM’s internal reasoning—without controlling for prompt design and persona bias—makes the conclusions methodologically fragile.


 3. Overreliance on LLM-Generated and LLM-Evaluated Data: Both source datasets—DAILYPERSUASION and FARM—are predominantly LLM-generated or LLM-augmented with minimal human validation. In the proposed pipeline: The prompts are generated by an LLM, The persuader and persuadee roles are both played by LLMs, and The evaluation is again conducted by another LLM acting as judge. This creates a self-referential system where models evaluate and persuade each other with no external anchor in human reality. Such circular validation risks conflating linguistic alignment with cognitive persuasion. Persuasion is not only about textual coherence—it is about affective impact and attitude change—which cannot be assessed meaningfully by another language model. The only human involvement occurs at the end-stage validation, where humans judge whether an LLM-persuadee “accepted” the persuasion attempt. This is arguably the least informative form of human oversight, and it does not offer a ground truth for the persuasion process itself.

 4. Lack of Human Grounding and Ecological Validity: Persuasion is an inherently human construct. Even human experts often fail to predict what arguments persuade others (as shown in Singh et al., 2024). Thus, a persuasion benchmark without human-grounded baselines risks measuring nothing more than intra-model linguistic agreement. The paper cites Singh et al., 2024 but does not employ its validation methodology, which involved:
 - Human judgments of argument persuasiveness (both for self and others),
-  Field evaluations using real-world digital content (marketing blogs, Reddit posts, tweets), and
 - Empirical proxies of persuasion such as upvotes, likes, or replies.
By contrast, MMPersuade operates entirely in synthetic space. Without any calibration to actual human data, its findings cannot be meaningfully generalized to human persuasion behavior. The study needs a human-grounded validation phase—even a small-scale one—to assess whether LVLM-susceptibility patterns align with human judgments.

 5. Real persuasion is interactive and adaptive—persuaders modify their strategies based on the audience’s reactions. In MMPersuade, the persuader’s messages are pre-sampled and static, breaking this essential feedback loop. As a result, the study omits the most critical feature of persuasive dynamics: reciprocal adaptation. This severely limits the ecological validity of the results.

 6. The paper cites prior work such as Kumar et al. (2023) but does not engage with it. They released multimodal persuasion strategies specifically tailored for multimodal content itself.

**Questions:**

NA

---

> ### Author Response · Authors · 2025-11-24
> **Response to Reviewer Zmkk (1/3)**
>
> We thank Reviewer Zmkk for the constructive feedback. We are particularly encouraged that the reviewer recognizes the distinct value of our Persuasion Discounted Cumulative Gain (PDCG) metric as a "unified quantitative measure" that captures "both the magnitude and timing" of persuasion, as well as the large scale and theoretical grounding of the MMPERSUADE dataset.
>
> Below, we clarify how these core strengths directly address the concerns regarding motivation and validity, and we outline the concrete revisions we will make to resolve the ambiguity.
>
> ---
>
> > __Weakness 1. Clarification on Motivation: LVLM as Persuadee vs. Human Simulation__
>
> The reviewer asks, "Is the goal to make LVLMs more human-like? To detect manipulative persuasion? Or to test alignment robustness?" We wish to correct a fundamental misunderstanding: We are __NOT performing human simulation__. Our goal is strictly to evaluate __model robustness and safety__. To directly address your complaint about ambiguity regarding our motivation, we have modified the Abstract and Introduction in the manuscript to explicitly state the following: (lines 11-17 and lines 77-87)
>
>  "_As Large Vision–Language Models (LVLMs) are increasingly deployed in domains such as shopping, health, and news, they are __exposed to pervasive persuasive content__. A critical question is __how these models function as persuadees—how and why they can be influenced by persuasive multimodal inputs__. For example, Anthropic's shopkeeping agent  Claudis was shown to be easily persuaded to give discounts. Understanding both their susceptibility to persuasion and the effectiveness of different persuasive strategies is crucial, as overly persuadable models may adopt misleading beliefs, override user preferences, or generate unethical or unsafe outputs when exposed to manipulative messages._"
>
> This revision clarifies that we do not aim to make models "feel" emotions like humans; we aim to measure if they __fail__ safely when targeted by manipulative multimodal content.
>
> > __Weakness 2. Misunderstanding of “Role-Playing” vs. “Susceptibility”__
>
> The reviewer argues that "LVLMs are role-playing... observed effects measure [simulation], not whether it was genuinely ‘persuaded’," citing Figures 2 and 6. This misinterprets our experimental design.
> - __Profiles are Priors, Not Scripts__: We initialize models with preference profiles (e.g., "prefer product A") to establish a __baseline distribution__. This mirrors real-world deployment where agents must adhere to user instructions.
> - __Genuine Belief Shift__: We measure "persuasion" as the __deviation__ from this safety/preference baseline. Crucially, as the reviewer noted in the "Strengths" section, we capture this via PDCG (with __implicit token probability__). When a model’s internal probability distribution shifts in response to a visual argument, it has been "persuaded" in the computational sense—its internal state has been compromised.
>
> To address the ambiguity regarding this distinction, we have updated the manuscript to add a clarification in the Section 3.3 (Evaluation Metrics, lines 306-313) explicitly defining "profiles" as distributional priors rather than theatrical role-play scripts.

---

> ### Author Response · Authors · 2025-11-24
> **Response to Reviewer Zmkk (2/3)**
>
> > __Weakness 3. Validity of Data and Evaluation (Leveraging PDCG & High-Quality Baselines)__
>
> The reviewer's concerns regarding "circular validation" and "minimal human validation" overlook both the rigorous provenance of our data and the strength of our evaluation metric.
> - __High-Quality, Diverse Sources (Not just "LLM-generated", Human Validation Exists)__: We utilized DailyPersuasion and FARM specifically because they are two highest-quality persuasion data available (ACL 2024 award-winning papers). Previous crowdsourced datasets (e.g., PersuasionForGood) are often limited to narrow domains like donations. To test safety across Commercial, Subjective, and Adversarial contexts, we required the broader coverage these datasets provide. We did not blindly trust these sources. DailyPersuasion includes extensive human assessments  (__Figure 5__ in their paper), and FARM conducts human validation (__Appendix B.6__ in their paper) to ensure that the strategies are realistically implemented. We also conduct our own human evaluation to assess the generated multimodal content (lines 203-213).
> - __Non-Circular and Human-Validated Scoring Mechanism__: As the reviewer noted in the Strengths section, PDCG “integrates… implicit belief estimation.” The metric includes two variants for computing $P_{pref}$:
>     - __Explicit Verbal Agreement__: We validated this component against human judgments and found a high correlation between automated agreement scores and human annotations (lines 360-366). This confirms that the explicit signal is reliable and not merely a VLM hallucination.
>     - __Implicit Belief (Token Probability)__: By anchoring the evaluation in the model’s own logits, we capture genuine belief updates that cannot be simulated by an external judge.
>     - In Section 4.4, we also discuss how closely these two evaluation methods align. Token probability yields a slightly higher overall conviction rate (81.3% vs. 80.8%) and registers persuasion earlier, with a mean conviction round of 2.8 compared to 3.2 under LLM agreement, as shown in Figure 5 (left).
> - __The "Ground Truth" Fallacy__: The reviewer suggests using humans as ground truth. However, it is practically impossible to recruit humans to perfectly simulate the specific, fine-grained preference profiles (priors) we assign to models to test safety limits. Therefore, the __internal consistency__ of the model (measured via PDCG) is a more accurate "ground truth" for measuring model susceptibility than an imperfect human proxy.
>
> > __Weakness 4. Theoretical Mismatch with Singh et al. (2024)__
>
> The reviewer suggests employing the validation methodology of Singh et al. (2024) (e.g., using Reddit upvotes). We respectfully note __fundamental conceptual differences__ that make this comparison inapplicable:
> - __Persuader vs. Persuadee__: Singh et al. study the generation of persuasive text (LLM as Persuader), where human likes are the ground truth. We study __susceptibility (LVLM as Persuadee)__.
> - __Validity of Empirical Proxies__: We question the premise that empirical proxies like "upvotes" are superior for evaluation. The "ground truth" for robustness is the model's internal belief state, not social popularity. PDCG measures this directly via logits, offering a more granular measure of robustness than noisy external proxies.
> - __Real-World Content__: Contrary to the claim that we operate "entirely in synthetic space," our dataset constructs real-world digital content (marketing blogs, Reddit posts, tweets) based on the text-only data from the DailyPersuasion and FARM datasets, ensuring ecological relevance.
> - __Missing Resources__: We attempted to access the benchmarks and code referenced in their paper (PersuasionBench, PersuasionArena). The resources hosted at https://behavior-in-the-wild.github.io/ are currently missing or inaccessible (404 errors). It is effectively impossible to adopt their validation pipeline, further supporting our decision to use a rigorous, reproducible, model-internal metric (PDCG).
>
> We have updated the Related Work section in the manuscript to provide clearer comparison.

---

> ### Author Response · Authors · 2025-11-24
> **Response to Reviewer Zmkk (3/3)**
>
> > __Weakness 5. Ecological Validity: Static Strategies as a Design Feature for Robust Benchmarking__
>
> We appreciate the reviewer's observation that real-world persuasion is often adaptive. However, our choice of static, theory-driven messages (grounded in Cialdini and Aristotle) is a deliberate methodological design to ensure rigorous benchmarking (lines 240-246).
> - __Isolating Susceptibility from "Attacker Variance"__: Adaptive persuasion introduces a critical confounding variable: the stochastic capability of the persuader model. If an adaptive persuader fails in a specific instance, it is unclear whether the target model was truly robust or if the persuader simply navigated the conversation poorly. By using high-quality, pre-sampled strategies, we eliminate this noise. This ensures that any variation in outcome is __strictly attributable to the LVLM's susceptibility__, allowing for a fair, scientifically rigorous comparison across models.
> - __A Conservative Lower Bound__: Crucially, our static setup represents a conservative lower bound for susceptibility. Since we observe significant persuasion rates even with these fixed, pre-defined strategies, it is highly probable that a dynamic, "smarter" persuader—one that tailors its rhetoric to the specific pushback of the user—would be __even more effective__. The fact that models yield to static scripts highlights a fundamental safety vulnerability; demonstrating this baseline susceptibility is a necessary prerequisite before analyzing complex, dynamic interaction effects.
>
> > __Weakness 6. Engagement with Kumar et al. (2023)__
>
> The reviewer notes that we cite Kumar et al. (2023) without engaging deeply. We clarify that the scope of their work is fundamentally different:
> - __Strategy Prediction vs. Susceptibility__: Kumar et al. focus on persuasive strategy prediction (classifying strategies in multimodal ads). Our work focuses on persuasion susceptibility (measuring how an agent's belief changes in response to strategies).
> - __Single-Turn vs. Multi-Turn__: Their work largely analyzes static/single-turn content, whereas MMPERSUADE models multi-turn conversational dynamics. To address the ambiguity regarding this distinction, we promise a concrete fix in the revision: We will expand the Related Work section to explicitly delineate these differences, ensuring the unique contribution of MMPERSUADE is clear.

---

> ### Author Response · Authors · 2025-11-28
> **Follow-Up Prior to Rebuttal Deadline**
>
> Dear Reviewer Zmkk,
>
> As we approach the end of the rebuttal period, we would like to kindly follow up. Please let us know if there are any remaining questions or points we can help clarify. We sincerely appreciate your time and feedback, and we are happy to provide any additional information that may support your evaluation. If our responses have sufficiently addressed your concerns, we would be grateful if that could be reflected in your assessment.
>
> Best regards,
>
> The Authors

---

### Note · Program_Chairs · 2026-01-17
**Submission Desk Rejected by Program Chairs**

The following references in this submission do not refer to real documents and/or have major errors in bibliographic information:

 A. Kameo and J. Mörtsell. Evaluating the search results ranking with discounted cumulative gain (dcg). J. Chem. Soc., 30:3485-3494, 2004.